# Surface, Interface, and Temperature Effects on the Phase Separation and Nanoparticle Self Assembly of Bi-Metallic Ni0.5Ag0.5: A Molecular Dynamics Study

**DOI:** 10.3390/nano9071040

**Published:** 2019-07-21

**Authors:** Ryan H. Allaire, Abhijeet Dhakane, Reece Emery, P. Ganesh, Philip D. Rack, Lou Kondic, Linda Cummings, Miguel Fuentes-Cabrera

**Affiliations:** 1Department of Mathematical Sciences, New Jersey Institute of Technology, Newark, NJ 07102, USA; 2Center for Nanophase Materials Sciences, Oak Ridge National Laboratory, Oak Ridge, TN 37831, USA; 3Department of Materials Science and Engineering, The University of Tennessee, Knoxville, TN 37996, USA

**Keywords:** molecular dynamics simulations, phase separation, metallic nanoparticles, self-assembly, core-shell nanoparticles

## Abstract

Classical molecular dynamics (MD) simulations were used to investigate how free surfaces, as well as supporting substrates, affect phase separation in a NiAg alloy. Bulk samples, droplets, and droplets deposited on a graphene substrate were investigated at temperatures that spanned regions of interest in the bulk NiAg phase diagram, i.e., miscible and immiscible liquid, liquid-crystal, and crystal-crystal regions. Using MD simulations to cool down a bulk sample from 3000 K to 800 K, it was found that phase separation below 2400 K takes place in agreement with the phase diagram. When free surface effects were introduced, phase separation was accompanied by a core-shell transformation: spherical droplets created from the bulk samples became core-shell nanoparticles with a shell made mostly of Ag atoms and a core made of Ni atoms. When such droplets were deposited on a graphene substrate, the phase separation was accompanied by Ni layering at the graphene interface and Ag at the vacuum interface. Thus, it should be possible to create NiAg core-shell and layer-like nanostructures by quenching liquid NiAg samples on tailored substrates. Furthermore, interesting bimetallic nanoparticle morphologies might be tuned via control of the surface and interface energies and chemical instabilities of the system.

## 1. Introduction

Recently, pulsed-laser-induced dewetting (PLiD) has been used to organize nanoparticles on surfaces with a correlated length scale. The PLiD exposes an ~10 ns pulsed laser to a metal thin film (single digits to tens of nm thick), which liquefies the film for up to tens of nanoseconds. During the liquid lifetime, the film [1,2,3,4] or lithographically pattered nanostructure [5,6,7,8,9,10,11,12,13] experiences instabilities. The balance of viscous, capillary, and inertial forces induces liquid phase transport at the nanoscale. Natural two-dimensional thin film (spinodal and nucleation) instabilities and one-dimensional Rayleigh–Plateau instabilities have been studied. Since the rapid solidification of the features locks in even metastable morphologies, the sequence of low laser fluence/low liquid lifetime pulse has revealed a transient behavior. While much of the work has been dedicated to elemental metals, multifunctional nanoparticles can be realized by exploiting competing chemical instabilities. For instance, metallic alloys with liquid and solid phase miscibility [13,14] /immiscibility [15,16] gap can lead to tunable/multifunctional nanoparticles, respectively. Beyond experimental studies, complementary continuum modeling [10,17,18,19] and molecular dynamics simulations [20,21,22,23] have been used to elucidate the various liquid phase instabilities and transport behavior operative in nanoscale metallic liquids. While historically mainly elemental films have been studied, we are turning our attention to alloys where competing chemical instabilities may also be operative during fluid mechanical evolution.

In order to study the evolution of a liquid alloy to create nanoparticles, one must consider three effects. First, the chemical composition of the alloy, which might lead to phase separation in certain temperature ranges. Second, the surface energies of the metals involved, as one expects that the metal with a smaller surface energy would migrate to the free surface. And third, the interaction of the alloy with the substrate that supports the liquid, which determines the wetting/dewetting angle and also can induce preferential migration of the lower interfacial energy liquid. Cumulatively, various nanoparticle morphologies can emerge depending on the chemical and surface/interface energies.

In this study, in order to understand these three effects, we investigate the Ni0.5Ag0.5 alloy. At the Ni0.5Ag0.5 atomic composition, the NiAg phase diagram contains four distinct regions: (i) Above ~2700 K, a liquid region phase where both Ni and Ag are miscible; (ii) between ~2700–1800 K, a liquid-liquid phase where Ni and Ag have limited solubility and two liquid phases emerge; (iii) between ~1700–1200 K, a liquid-solid phase where the Ag-rich phase is liquid, the Ni-rich phase is crystalline and both have limited solubility; and finally (iv) below ~1200 K, a solid-solid phase where both Ni-rich and Ag-rich phases are crystalline and again have limited solid solubility. The phase fraction and specific phase compositions, of course, vary with temperature.

Here, we use classical molecular dynamics (MD) simulations to study the Ni0.5Ag0.5 chemical composition, and we focus on how surface and liquid-substrate interfacial interactions affect phase separation at the aforementioned regions of interest in the phase diagram. The results obtained provide a road map for future studies, which will investigate competing chemical and hydrodynamic instabilities that occur during the bimetallic liquid phase assembly of nanoparticles.

## 2. Materials and Methods

The simulations started from a 256 atom structure of Ni0.5Ag0.5, created from a face-centered cubic (FCC) lattice, where Ni and Ag were randomly mixed and the lattice parameter of Ni (3.524 Å) was assumed in the original structure. Subsequent to generating the Ni0.5Ag0.5 lattice, its total energy was minimized. An illustration of this structure is shown in Figure 1.

The 256 atom NiAg structure was then expanded in the x, y, and z directions to generate a sample that contained 55,296 atoms. We refer to this sample as the bulk sample, as we employed periodic boundary conditions at each +/− x, y, and z boundary. Then, the bulk sample was studied, first assuming the isothermal-isobaric (NPT) ensemble for 300 ps, followed by a canonical (NVT) ensemble for 600 ps, followed by the microcanonical ensemble (NVE) for 300 ps, all using a time step of 1 fs. These simulation times were found to be sufficient to converge the values of pressure, temperature, and energy in NPT, NVT, and NVE, respectively. The highest temperature considered was 3000 K, and once the sample was equilibrated with NVE at this temperature, it was quenched by reducing the temperature in 200 K increments until reaching 800 K. The corresponding atomic densities for the equilibrated 3000 K and 800 K structures were 54 and 65.8 atoms/nm^3^, respectively. Because at every temperature the sample was equilibrated for 1.2 ns (300 ps NPT, 600 ps NVT and 300 ps NVE), the cooling rate in our simulations was 200 K every 1.2 ns, i.e., 1.67 × 10^11^ K/s. The melting points of Ni and Ag were 1726 and 1235 K, respectively, and by creating a Ni0.5Ag0.5 sample at different temperatures we aimed to study the different regions that appeared in the phase diagram.

The embedded-atom method (EAM) potential derived by Zhou et al. [24] was used to describe the Ni-Ni, Ag-Ag, and Ni-Ag interactions. This potential was developed for studying a NiAg alloy and it is the only NiAg potential we know of that is capable of capturing the relevant Ni-Ag phase separation. Indeed, we used the universal form of the EAM potential for Ni and Ag, and the NiAg Finnis–Sinclair potential of Pan et al. [25]. With the former, no phase separation was observed when the system was similarly quenched; with the latter, we obtained a similar radial distribution function to that shown by Pan et al. Figure 7 of [25] for a Ag_80_Ni_20_ alloy. However, when we used this potential to quench Ni0.5Ag0.5 from 3000 K to 800 K with a cooling rate 1.67 × 10^11^ K/s, phase separation was not observed.

To ensure that the Zhou et al. [24] EAM potential for Ni0.5Ag0.5 was accurate for the individual elements, we melted and cooled down a sample of 2048 atoms of Ni and Ag using NPT with melting and cooling rates of 2 × 10^13^ K/s (in 100 K increments for 500 ps each). Figure 2 shows the change in volume with temperature for the samples containing only Ni and only Ag, respectively. A sudden increase/decrease in the volume indicates melting/freezing has taken place and the hysteretic behavior is consistent with what is commonly observed [26]. In the case of Ni (Ag), the volume increases suddenly between 1800 K and 1900 K (1300 and 1400 K), which is close to the experimental melting point of 1726 K (1235 K). Upon cooling, the Ni (Ag) volume decreases dramatically at a temperature between 1000 and 900 K (800 and 700 K). Table 1 shows the slopes of the plots during the heating and cooling. Ag has a higher dV/dT relative to Ni, which is consistent with the fact that Ag (~19 × 10^−6^/K) has a higher coefficient of thermal expansion than Ni (~13 × 10^−6^/K). As expected, both liquids have higher dV/dT than their respective solids. For comparison, we also heated and cooled a sample of 2048 atoms of Ni0.5Ag0.5 atomic composition; the results are also shown in Figure 2.

In this case, upon heating (cooling), only one abrupt volume change was observed between 1000 and 1100 K (900 and 800 K). This abrupt change was due to the Ag phase transformation, where both the heating and cooling were shifted to slightly lower temperatures, which could have been due to the smaller cluster size of the Ag. The slope of the cooling curve of the Ni0.5Ag0.5 is approximately the average between the Ag and Ni cooling curve slopes. Notably, the Ni phase transformation is not observed, which is likely due to the sluggish phase separation and perhaps supersaturation of the Ni phase. Interestingly, the Ni0.5Ag0.5 slope is also close to the average of the solid Ni and liquid Ag (2.8 ${Å}^{3}/K$). Interestingly, the slope of the Ni0.5Ag0.5 heating curve is close to that of pure Ni and lower than the average.

From the bulk sample created at each temperature, we generated droplets by simply adding a vacuum interface. It was found, then, that running 1800 ps of NVT and 300 ps of NVE was enough to equilibrate the resultant droplets. An example of an equilibrated NiAg droplet at 2000 K is shown in Figure 3a. Finally, a droplet at 2000 K was deposited on a single layer graphene substrate at an initial distance of 3 Å, see Figure 3b. The droplet was subsequently equilibrated using 1500 ps of NVT, while the substrate, as in previous studies [27], was kept frozen.

When the droplet was deposited on the graphitic substrate, the metal-C interactions were described with a 12-6 Lennard-Jones potential given by:(1)$$V(r)=4\varepsilon \left[{\left(\frac{\sigma }{r}\right)}^{12}-{\left(\frac{\sigma }{r}\right)}^{6}\right],r<r_{c}$$
where ϵ is the depth of the potential well, σ is the distance at which the potential is zero, and $r_{c}$ is the truncation radius. Previous studies [27,28,29,30,31] have provided values for $\epsilon ,\sigma ,\;\mathrm{and}\;r_{c}$ but, as explained in Appendix A, we found that none of these sets of values were able to reproduce the contact angle of pure Ni and Ag liquid droplets deposited on graphite. Here, we find that using the values in Table 2 for $\epsilon ,\sigma ,\;\mathrm{and}\;r_{c}$, respectively, we obtain a contact angle of $59^{\circ }$ for Ni on graphite, and $145^{\circ }$ for Ag on graphite; these theoretical contact angles are very close to the values found experimentally (Ag-C = $135^{\circ }$ and Ni-C = $60^{\circ }$) [32,33,34,35,36,37]. All the simulations were done with the software LAMMPS [38].

## 3. Results

In what follows, we show how temperature and environment (bulk, suspended droplet, or droplet on graphite), affect the phase separation and nanostructure morphology.

### 3.1. Bulk Samples

Crystallization and phase separation are realized in MD simulations by calculating the radial pair distribution function (RDF). Figure 4 shows the RDF (computed with OVITO [39]) for the Ni0.5Ag0.5 system at all the temperatures considered. Each panel in the figure shows the RDF of Ni-Ni, Ag-Ag, and Ni-Ag. At 3000 K, the RDF shows that there is a slight preference to form homogenous pairs, i.e., Ni-Ni and Ag-Ag, rather than heterogeneous Ni-Ag pairs. The difference is slight, and it can be said that at this temperature the system is a miscible liquid of Ni and Ag. According to the phase diagram, the onset of phase separation starts below 2700 K. In the simulations, phase separation starts clearly at 2400 K. As seen in Figure 4, at 2400 K, the first Ni-Ni and Ag-Ag peaks increase while the first peak for Ni-Ag decreases, a sign that Ni and Ag are forming homogenous clusters and that heterogeneous clusters containing Ni-Ag are becoming smaller and less numerous. This trend continues down to 2000 K, and the fact that between 2400 and 2000 K there are no clear second and third peaks in the RDFs indicates that the system is still liquid, albeit immiscible. At 1800 K, Ni is close to its melting point and the first peak of the RDF has increased considerably, while a second peak has emerged. Ag, on the other hand, still remains liquid at 1800 K. Between 1600 and 1400 K, the crystallization of Ni is obvious, Ag still remains liquid, and the number of Ni-Ag pairs has decreased even further. The system is now phase separated into a Ni-rich crystal and an Ag-rich liquid. At 1200 K and below, the Ag-rich phase has already started to crystallize, and the system consists of a mixture of Ag-rich and Ni-rich crystalline phases, where both phases have very low solubility of the other constituent. The amplitudes of the first peaks are plotted in Figure 4d, showing an increase in pure metal pairs (Ni-Ni, Ag-Ag) and a decrease in mixed pairs (Ni-Ag) with decreasing temperature.

Phase separation is also observed with the coordination number, CN. The CN of the bulk samples at different temperatures is shown in Figure 5. Here, the CN was computed using the Visual Molecular Dynamics (VMD) software [40], by prescribing the radius at which the RDF attains the first minimum, corresponding to the first coordination number, and was performed for each pure and mixed pair. At 3000 K, the number of Ag (Ni) neighbors around Ag (Ni) is 8 (6.2), whereas the number of Ag (Ni) neighbors around Ni (Ag) is 5.8. Upon cooling from 3000 K, the CN remains constant until about 2400 K, when the number of Ag (Ni) neighbors around Ag (Ni) starts to increase slightly, while the number of Ag neighbors around Ni starts to decrease, also slightly. Below 2000 K, the rate of change of the CN increases and there is a sharp increase and decrease in the number of homogenous and heterogeneous pairs, respectively. At 800 K, there are very few heterogeneous pairs while the homogenous ones have reached a value of 12 in the CN, which is consistent with the FCC crystal structure.

Despite the fact that the cooling rate used here is much greater than the rates used in typical PLiD experiments, the MD simulations with the atomic potential are still capable of capturing phase separation in Ni0.5Ag0.5, in accordance with the experimental phase diagram. This encouraged us to explore the effect of a free surface and a supporting graphene substrate on phase separation.

### 3.2. Droplets

To investigate the free surface effects on phase separation, we added a vacuum interface to each of the already extant bulk samples and equilibrated each resultant droplet. Equilibration was achieved with 1800 ps of NVT. This approach, as compared to directly quenching a single droplet from 3000 to 800 K (which was avoided due to surface evaporation of Ag atoms), reduces the effect of Ag surface migration, as each bulk sample starts from a more nanogranular initial condition. Nonetheless, we expect that the results obtained in this way approximately represent the effect of Ag surface migration in the phase separation of a Ni0.5Ag0.5 droplet. As it will be seen, even at low temperature, where diffusion is slower, we observe the expected Ag diffusion towards the surface of the droplet.

The RDFs for the droplets, along with their peak amplitudes, at all temperatures studied are shown in Figure 6. These RDFs are similar to those of the corresponding bulk samples, see Figure 4, and thus at first one might conclude that phase separation is not significantly affected by the presence of a free surface. However, as Figure 7a,b illustrates, at 2200 K the NiAg droplet’s surface is preferentially Ag-rich due to its lower surface energy; for instance, the surface energies of Ni and Ag at their respective melting temperatures are approximately 1.78 N/m and 0.93 N/m. To demonstrate the morphology evolution, Figure 7c shows the plots of the relative Ag and Ni concentration in 5 Å concentric annuli slices as a function of the inner radius of the slice. We did not consider spheres beyond an inner radius of 60 Å, as any atoms at locations beyond this radius are either due to small perturbations in the spherical shape or due to evaporated particles.

At 3000 K, the local distributions of Ni and Ag are nearly equal, with a slightly higher concentration of Ag at the surface as well as preferential Ag evaporation, see Figure 7c. Except for the surface, the amounts of Ni and Ag are practically the same everywhere in the droplet. This, together with the corresponding RDFs, indicates that the system is not phase separated, i.e., is a miscible liquid. As the temperature decreases, the concentration of Ag atoms in the surface increases steadily, and similarly to the bulk simulation RDFs, phase separation is initiated at ~2400 K (the atomic local distribution analysis for all the temperatures in this study is shown in the Appendix A). At 1800 K, close to the Ni melting point, the following significant change is observed: the concentration of Ni (Ag) increases (decreases) significantly in the middle of the droplet (i.e., the region between the surface and the center of the droplet), whereas, the opposite effect is seen in the center. To understand this behavior, Figure 8a–c shows a cross section of the droplet at 2000, 1800, and 1600 K.

As shown in Figure 8, it is observed that Ni clustering is clearly occurring at 2000 K, and that at 1800 K the Ni grains coarsen and occupy the middle section of the droplet; at 1600 K, the Ni solidifies as evidenced by the RDF peak increase and coarsens nearly to a single large cluster with a few Ag cluster inclusions. At the liquid-to-solid phase transformation, the solubility of Ag in Ni also drops. Finally, at 1600 K the Ni cluster occupies most of the interior of the droplet, whereas, the Ag atoms migrate to the surface and form a shell around the Ni core. As seen in Figure 8d, the core-shell morphology continues down to 800 K; at this temperature, the Ni cluster is displaced from the sphere centroid, but the surface layer of Ag is still present. Notably the solubilities at 800 K, at which both metals are in a solid state, is very low as evidenced by the few solute atoms in each solvent matrix.

### 3.3. Droplets on Graphite

An equilibrated droplet at a temperature of 2000 K was deposited on a one-layer graphene substrate and subsequently re-equilibrated. Next, the droplet was quenched to 1600 K with a cooling rate of 1.33 × 10^11^ K/s. Figure 9 shows snapshots of a cross section of the droplet on graphite at 2000, 1800, and 1600 K. As explained in the Methodology section, the Ni-C and Ag-C interactions were described with a Lennard–Jones potential adjusted to reproduce the wetting angles of liquid droplet Ni and Ag on graphite. This produces a Ni-C interaction (ϵ=0.072 eV) that is stronger than that for Ag-C (ϵ=0.01 eV). Consequently, when a droplet of Ni0.5Ag0.5 at 2000 K is deposited on graphite, Ni atoms migrate towards the C atoms, whereas, Ag atoms migrate to the surface of the droplet. This creates a layered-like structure in the Ni0.5Ag0.5 droplet, with Ni (Ag) occupying most of the graphite-metal (vacuum) interface, see Figure 9a. Lowering the temperature to 1800 K and then 1600 K (Figure 9b,c) does not change this migration of Ni and Ag. When the temperature decreases, the solubility decreases, and the coarsening of Ag and Ni takes place. However, because of the presence of a graphite substrate, Ni agglomeration is located mostly near the droplet-substrate interface. This is consistent with the fact that Ni has a lower surface energy than Ag on graphite, and thus the contact angle resembles that of Ni.

To make clear the layering effect seen in Figure 9, atomic compositions of Ni and Ag are plotted in Figure 10 as a function of the distance from the substrate. These slices were taken in 5 Å increments from the droplet-substrate interface to the top of the droplet. In each slice the Ni and Ag compositions were both measured. Figure 10 reveals that the crossover point where the slice compositions are equal are all at approximately 8 Å from the substrate. Below this point the composition of Ni is higher due to the lower surface energy of Ni-C relative to Ag-C. Interestingly, the wetting angle decreases with decreasing temperature as evidenced by the change in height for the composition profiles. The larger Ni-C interface at a lower temperature causes the total nickel content to be higher in this ~8 Å layer. Thus, as is illustrated in the graphs, the Ni composition increases above the crossover point with increasing temperature.

## 4. Conclusions

Molecular dynamics simulations were used to investigate the effects of free surface and substrate in the phase separation process of a NiAg alloy. It was found that the atomic potential employed in the simulations was capable of reproducing the phase separation observed in the experimental phase diagram. Subsequently, droplets were created, and it was found that while phase separation still occurred, surface effects drove Ag towards the surface of the droplet substrate while Ni moved towards the interior. This led to the creation of Ni-Ag core-shell nanodroplets, with Ni in the interior and Ag in the surface. On the other hand, when these droplets were deposited on a graphitic substrate, phase separation led to a layered-type structure in which Ni agglomerated close to the substrate, while Ag still migrated to the surface of the droplet.

## Figures and Tables

**Figure 1 nanomaterials-09-01040-f001:**
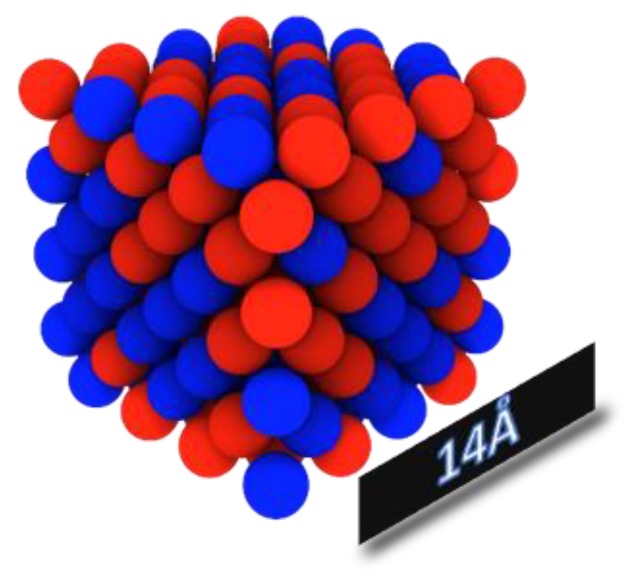
FCC structure of NiAg with 256 atoms and a 50/50 composition.

**Figure 2 nanomaterials-09-01040-f002:**
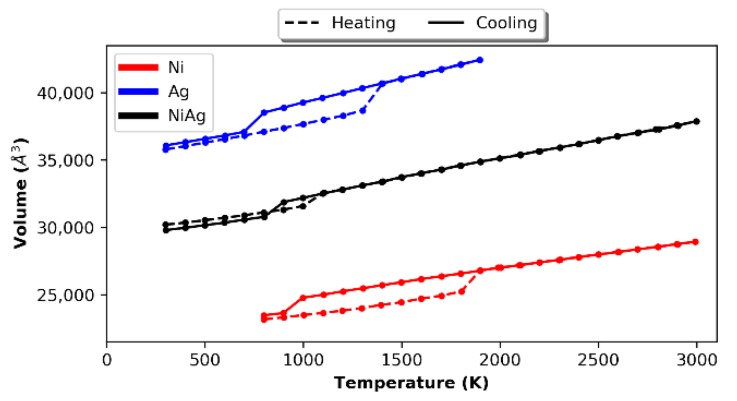
Melting and cooling of a 2048 atom sample of Ni (**red**), Ag (**blue**), and NiAg (**black**).

**Figure 3 nanomaterials-09-01040-f003:**
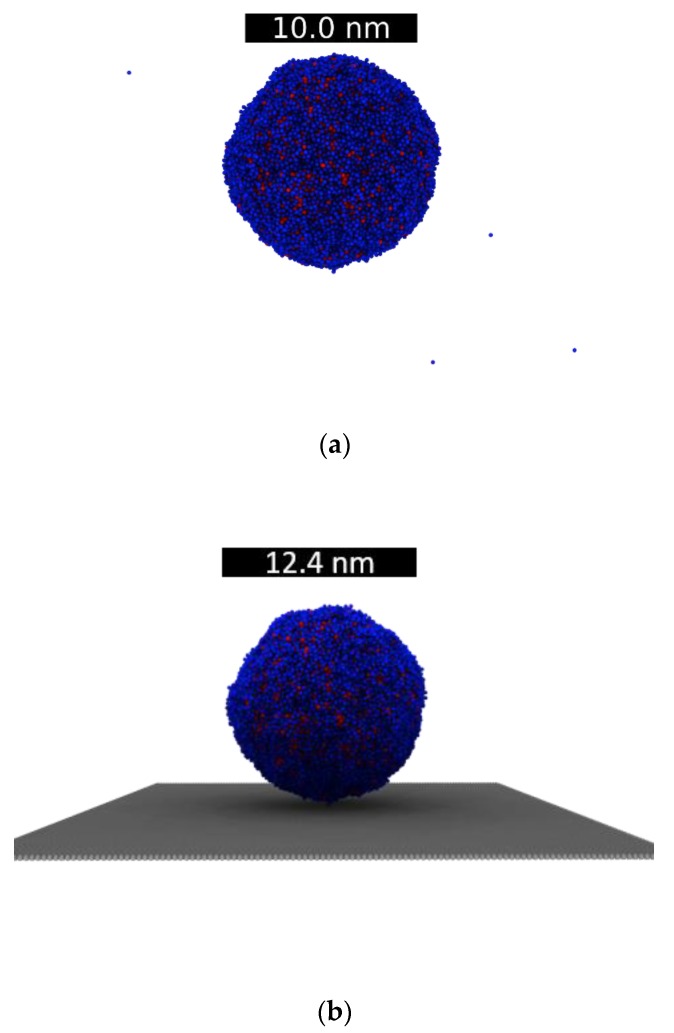
(**a**) Droplet of NiAg at 2000 K. (**b**) Droplet of NiAg at 2000 K deposited on 1-layer of graphite. The scale bar on (**b**) corresponds to the diameter of the droplet. Color code: Ni, red and Ag, blue.

**Figure 4 nanomaterials-09-01040-f004:**
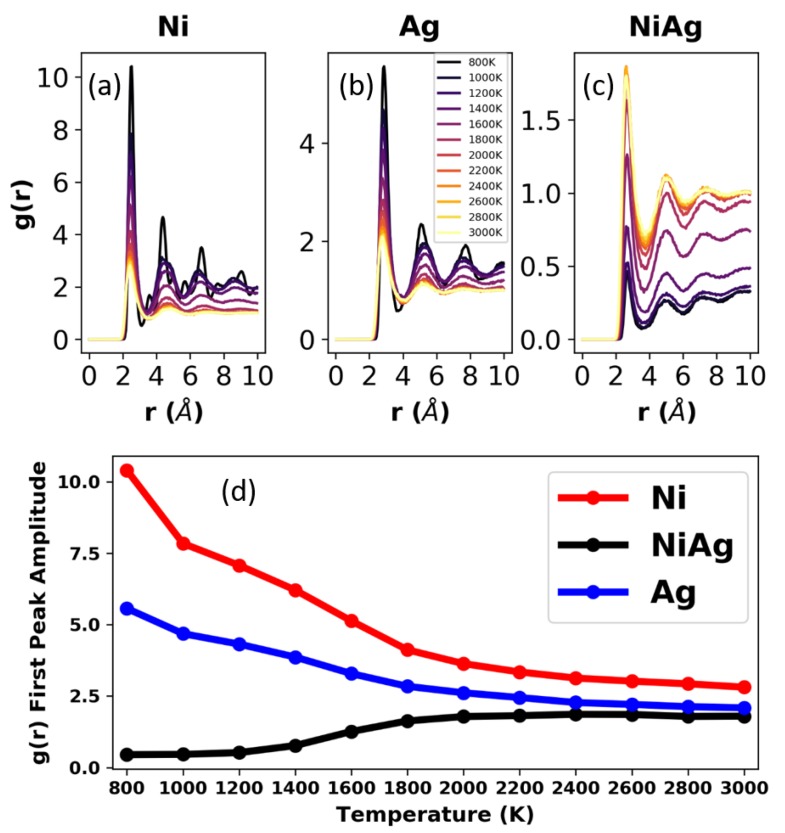
Radial distribution functions (RDFs) for the bulk samples at all the temperatures studied for Ni (**a**), Ag (**b**), and NiAg (**c**). (**d**) Plot of the the amplitude of the first peak (located between radii of 2 and 3 angstroms), as a function of temperature for Ni, Ag, and NiAg.

**Figure 5 nanomaterials-09-01040-f005:**
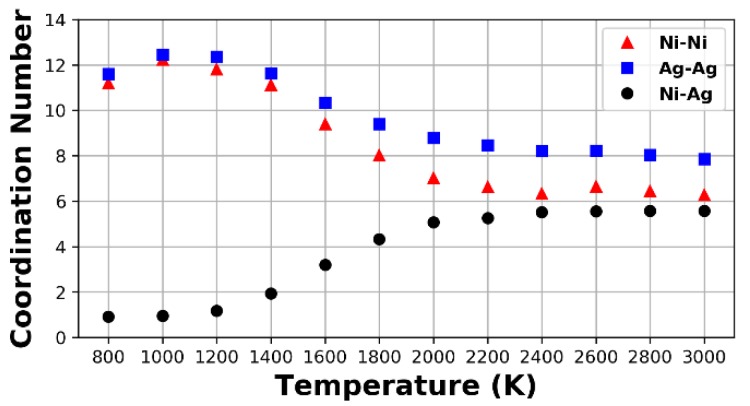
Coordination numbers for the bulk samples at different temperatures.

**Figure 6 nanomaterials-09-01040-f006:**
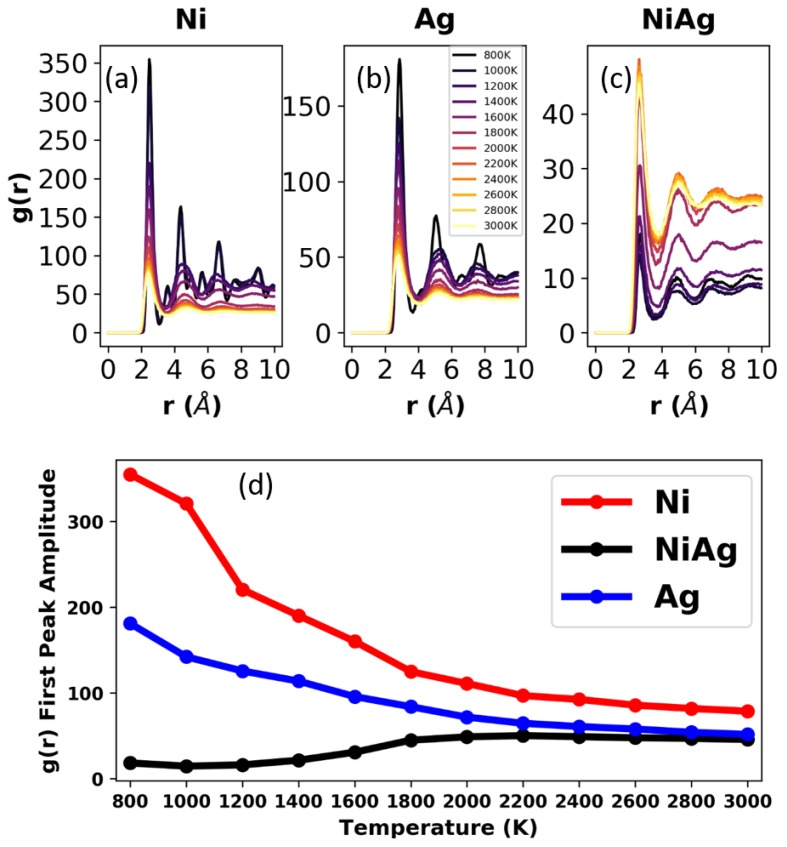
RDFs for the droplets at all temperatures for Ni (**a**), Ag (**b**), and NiAg (**c**). (**d**) Plot of the amplitude of the first peak (located between radii of 2 and 3 angstroms), as a function of temperature for Ni, Ag, and NiAg.

**Figure 7 nanomaterials-09-01040-f007:**
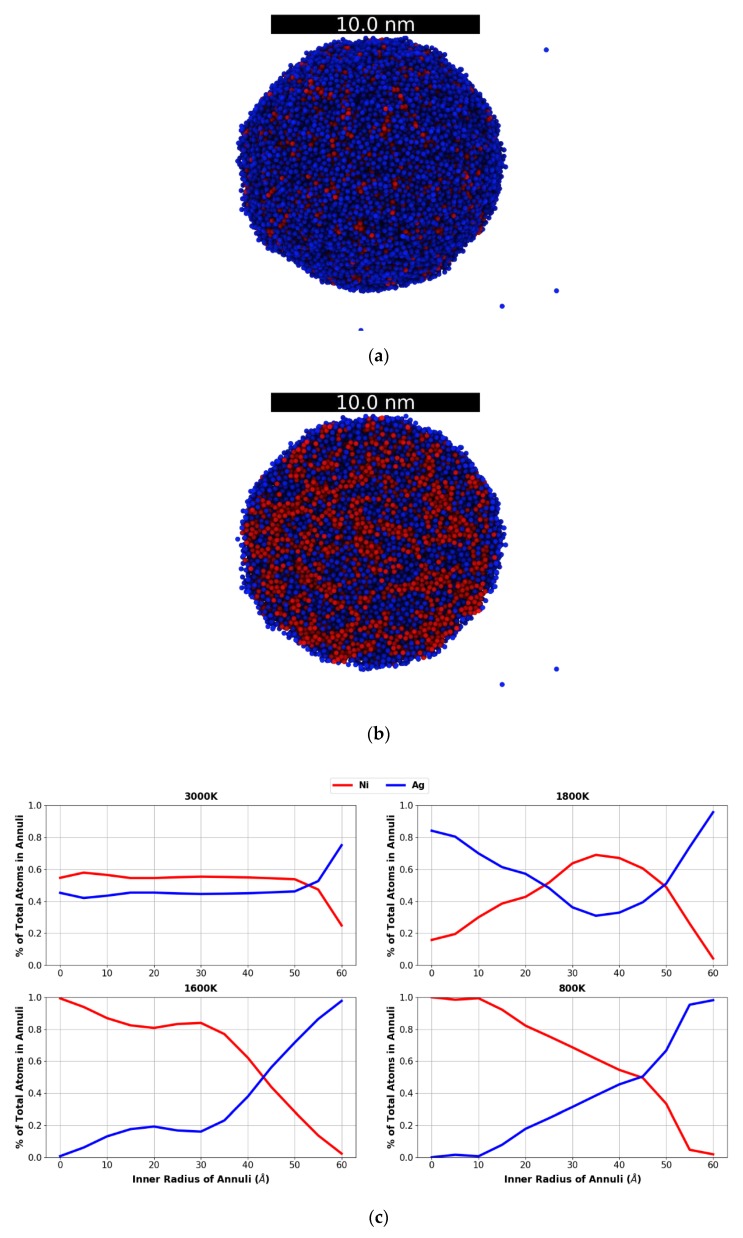
(**a**) NiAg droplet at 2200 K showing preferential movement of Ag to the surface, (**b**) slice of NiAg droplet at 2200 K, and (**c**) atomic concentration distribution analysis for the droplets at 3000 K, 1800 K, 1600 K, and 800 K. Color code: Ni, red and Ag, blue.

**Figure 8 nanomaterials-09-01040-f008:**
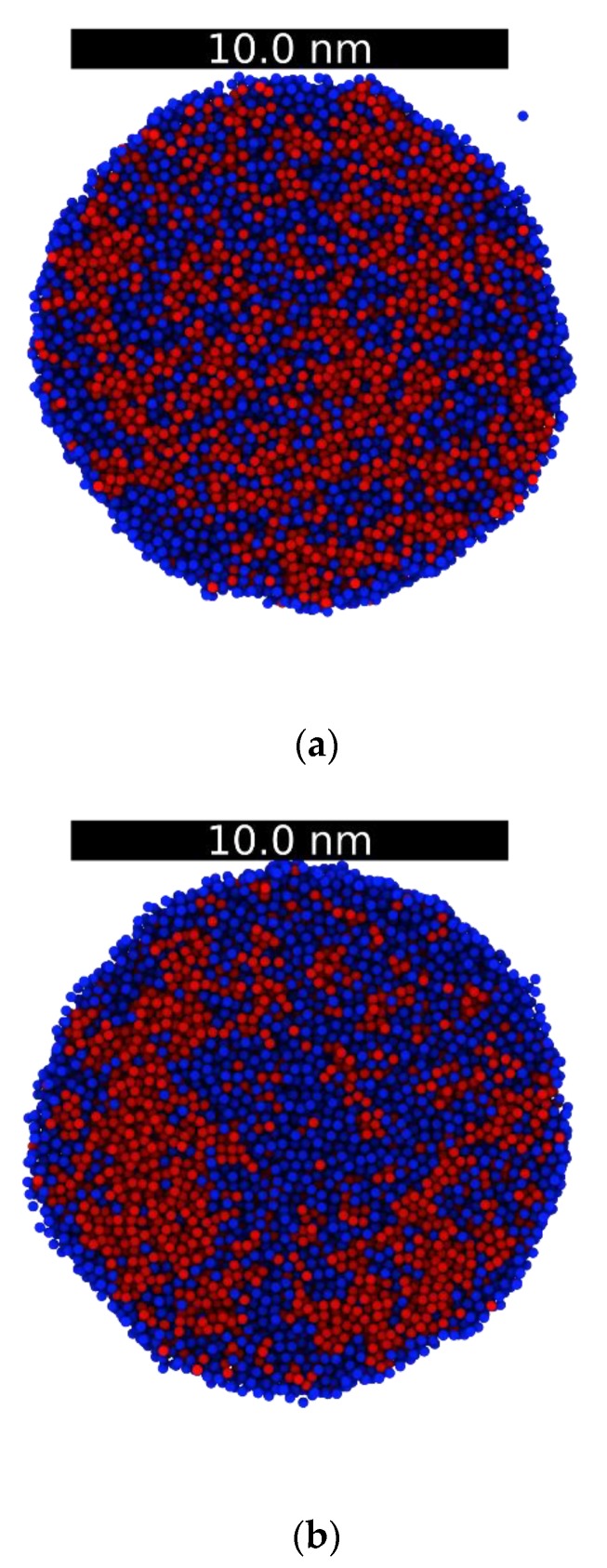
Cross sections of the droplets at 2000 K (**a**), 1800 K (**b**), 1600 K (**c**), and 800 K (**d**). Color code: Ni, red and Ag, blue.

**Figure 9 nanomaterials-09-01040-f009:**
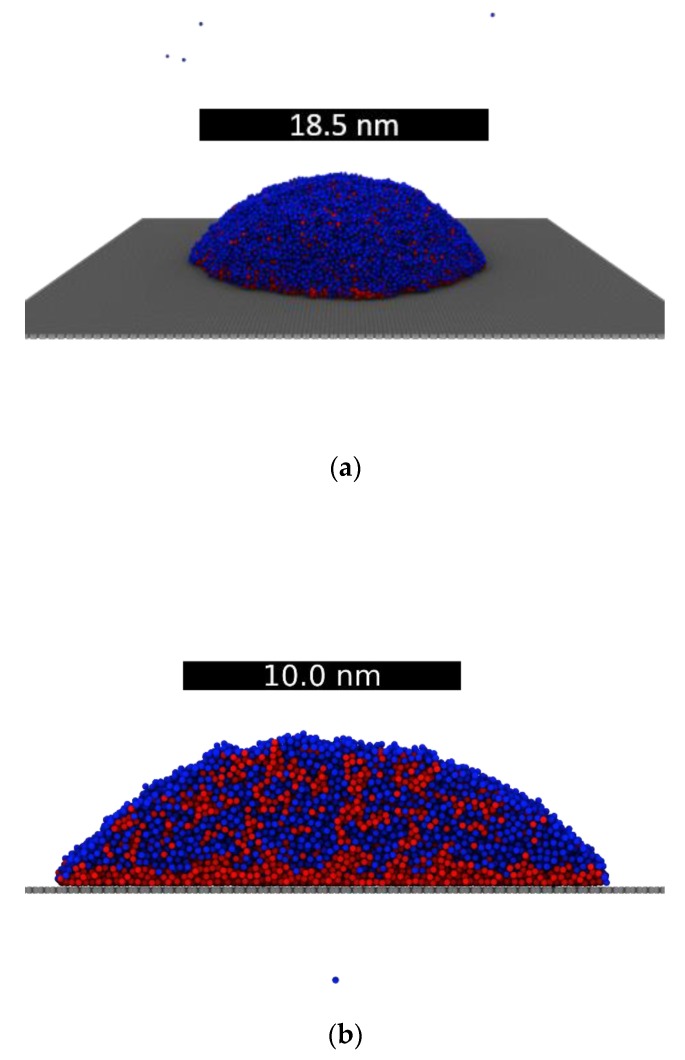
(**a**) 2000 K droplet deposited on one-layer of graphite. (**b**), (**c**), and (**d**) a cross-section snapshot at 2000, 1800, and 1600 K, respectively. The scale bar in (**a**) corresponds to the length of the droplet. Color code: Ni, red; Ag, blue; and C, grey.

**Figure 10 nanomaterials-09-01040-f010:**
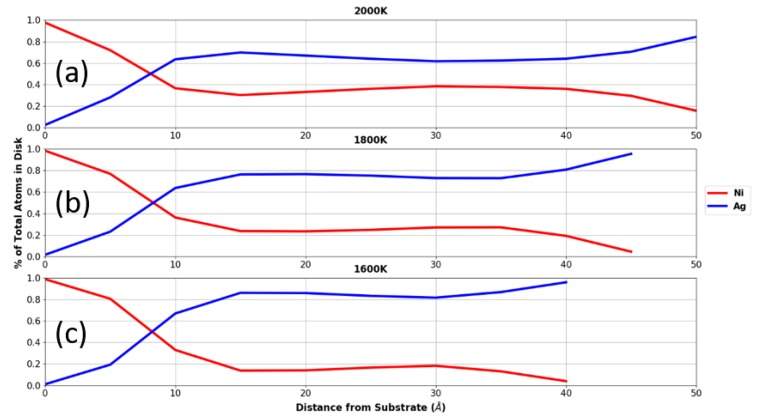
Atomic concentration distribution analysis of the droplets at (**a**) 2000 K, (**b**) 1800 K, and (**c**) 1600 K on substrates as a function of the distance from the substrate.

**Table 1 nanomaterials-09-01040-t001:** Slope of melting and cooling curves given in Figure 2 for Ni, Ag, and NiAg.

Element	$\mathrm{Solid}\;\mathrm{Phase}\;\mathrm{Slope}\;({Å}^{3}/K)$	$\mathrm{Liquid}\;\mathrm{Phase}\;\mathrm{Slope}\;({Å}^{3}/K)$
Ni	2.047	2.072
Ag	2.855	3.549
NiAg	1.962	2.815

**Table 2 nanomaterials-09-01040-t002:** Lennard-Jones parameters for Ni-C and Ag-C.

Interaction	ϵ (eV)	σ (Å)	$r_{c}\;\left(Å\right)$
Ni-C	0.072	2.8	11.0
Ag-C	0.01	3.006	11.0

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
