# Peer review of "Surface, Interface, and Temperature Effects on the Phase Separation and Nanoparticle Self Assembly of Bi-Metallic Ni0.5Ag0.5: A Molecular Dynamics Study"

_nanomaterials, 2019, doi:10.3390/nano9071040_

Round 1

Reviewer 1 Report

The authors have performed molecular dynamics simulations of mixtures of Ni and Ag as a solid phase, as a nano-droplet, and as a nano-droplet deposited on a graphite surface. They are able to reproduce the experimental melting and phase separation behaviour of the alloy in the solid phase. Additionally, they observe interesting phase separation phenomena in the nano-droplet in vacuum and the nano-droplet deposited onto a graphite surface. These depend on the relative values of the surface tension of the two metals with respect to vacuum and the graphite surface. This is an interesting study which would be useful to readers interested in physic-chemical aspects of metal alloy nanoparticles and for those interested in their practical uses. I would recommend the manuscript for publication in Nanomaterials once the following points have been addressed.

-As a matter of principle, the authors should give the functional form and parameters of the EAM potential they used in the simulations in the text or a Supporting Information section.

-In Table 2, are the Lennard-Jones parameters those for the Ni-C and Ag-C interactions? If so, the authors should state it more clearly.

-It would be useful if the authors provide a length scale marker on Fig. 3 and in other figures with snapshots of the nanosphere.

-Do the authors have the values of the surface tensions for Ni and Ag in any of the range of temperatures studied in this work? Do they have the contact angles for Ni and Ag on the graphite surface within the temperature range of the simulations? These data would make the interpretation of the results somewhat simpler from the outgo.

-For Fig. 9, can the authors perform a concentration profiling as a function of distance from the surface similar to what they have done in Fig. 7(c)? This will give a quantitative appreciation of the degree of layering seen for the deposited droplet.

-Fig. 2 shows a melting / freezing hysteresis loop often seen for finite systems. The authors should discuss this behaviour in a bit more detail. Do the authors see any melting / freezing hysteresis in the simulations of the larger samples they studied in this work? This should also be mentioned.

Reviewer 2 Report

In this manuscript by Allaire et al. the authors report results from their computational study on the effects of temperature on the phase separation in a bi-metallic Ni0.5Ag0.5 alloy in its bulk state as well as in a droplet in vacuum and on a graphite surface. The authors employ classical MD simulation to simulate the Ni-Ag alloy in the different environments mentioned above between the temperatures 3000 K to 800 K. This study shows that it is possible to make Ni-Ag core-shell  and layered nanostructures by controlling the interfacial environment. This work is a part of a larger endeavor by the authors to understand how hydrodynamics instabilities can be used to make metallic nanostructures. As such, I believe it should be of interest to the readers of 'Nanomaterials'. However, there are several issues listed below that should be addressed before this work can be accepted for publication and might need another round of review. 

1. The authors have included some details on the simulation and some results in a supplementary file which is not available. While this information is not of central importance to the paper, in absence of this information, reviewing becomes difficult. The supplementary file should be provided.

2. Some details about the simulation are missing from the Materials and Methods section. For example, what are the cell vectors of the FCC lattice made in the beginning? What are the sizes of the bulk system and the droplets (volume/density)? What calculation time step was used in the simulation? 

3. The V-T curves shown in Figure 2 are very smooth and suggest that perhaps a temperature ramp was used for these heating and cooling cycles. While LAMMPS, the package used by the authors ,does allow a temperature ramp to be set up, it is not very clear if it was used. This should be quite obvious to the reader, however, confusion ensues especially since in the first paragraph of page 3, the authors state that they cooled a different system by simulating it for 1.2 ns each at temperature intervals of 200 K thereby obtaining a cooling rate of 1.67*10^11 K/s. In this case, obviously cooling was not via a ramp.

4. It is very hard to follow the description of Figure 4 above it and its interpretation because of crowding by several curves with graded colors. It would help if a 4th plot can be added to this figure showing the intensity of the first RDF peaks of the three pairs as a function of temperature. This plot would be quite similar to Figure 5, but it will help in direct interpretation of Figure 4. A similar 4th plot can also be added to Figure 6. 

5. Although as the authors state, this is a part of their study to understand how hydrodynamics instabilities can be used to make metallic nanostructures, the connection between this work and the instabilities is not very clear. As the instabilities are not mentioned anywhere in the results section, in my opinion, in the current version of the manuscript, nothing will be lost if instabilities were not mentioned in the manuscript. 

Round 2

Reviewer 1 Report

The authors have responded to my comments and the paper is clearer now. I would recommend its publication in Nanomaterials.  

Reviewer 2 Report

The authors have satisfactorily responded to the reviewers' comments in the first round of review and have modified the manuscript accordingly. As a result of this, the work is now suitable for publication. I recommend accepting this manuscript for publication in it's current form.